# Robust changes in global subtropical circulation under greenhouse warming

Shijie Zhou[1], Ping Huang [1,2] ✉, Lin Wang [1,2] ✉, Kaiming Hu [1,2], Gang Huang [2,3] & Peng Hu[4]

The lower tropospheric subtropical circulation (SC) is characterized by monsoons and subtropical highs, playing an important role in global teleconnections and climate variability. The SC changes in a warmer climate are influenced by complex and region-specific mechanisms, resulting in uneven projections worldwide. Here, we present a method to quantify the overall intensity change in global SC, revealing a robust weakening across CMIP6 models. The weakening is primarily caused by global-mean surface warming, and partly counteracted by the direct $CO_2$ effect. The direct $CO_2$ effect is apparent in the transient response but is eventually dominated by the surface warming effect in a slow response. The distinct response timescales to global-mean warming and direct $CO_2$ radiative forcing can well explain the time-varying SC changes in other $CO_2$ emission scenarios. The declined SC implies a contracted monsoon range and drying at its boundary with arid regions under $CO_2$-induced global warming.

The subtropical circulation (SC) in the lower troposphere consists of subtropical highs over the oceans and monsoons[1]. The proportion of subtropical highs and monsoons varies seasonally, following the seasonal variation in the descending branches of the Hadley cell and the land–sea thermal contrast[2–5]. The SC connects the trade winds with the midlatitude westerlies and transports the tropical moisture poleward via the western flank of the subtropical high[6–9], playing an important role in global energy and moisture transport. Changes in intensity and location of the SC can impact the tropical cyclone tracks[10], modulate the distribution and variability of rainfall over East Asia, North and South America, and South Africa[11–14], and bring extreme events such as droughts and heatwaves[15,16].

Projection of the SC changes under greenhouse warming has been widely studied[17–22]. When the divergent SC is projected to weaken due to a weakened tropical overturning circulation under global warming[23–26], the change in the dominant rotational component of SC remains inconclusive[19,20,22,27–29]. The projection for the rotational SC represented by 850 hPa streamfunction (hereafter referred to as SC) is

crucially dependent on the location and season (Fig. 1 and Supplementary Fig. S1). The SC changes are distinct during the boreal summer and winter in the Northern Hemisphere, whereas the seasonal variation of the SC changes in the Southern Hemisphere is relatively weak. The typical seasonal SC changes include the slightly weakened Australian summer monsoon (Fig. 1a), the slightly strengthened East Asian summer monsoon, and the robust westward shifted North Atlantic subtropical high in JJA (Fig. 1b). On closer inspection, the SC change projection even depends on the chosen domain and metrics sensitively. The domain-average low-level streamfunction in the center of the subtropical high increases under global warming[20,29]. The eddy streamfunction in an extended domain exhibits an insignificant change in the North Pacific subtropical high[22], but several studies selecting variable-dependent polygonal domains project a weakened North Pacific subtropical high[19,28]. Furthermore, the responses of SC to global warming often exhibit a mixture of varying intensity and shifting location, such as the intensified and westward shifting North Atlantic subtropical high in boreal summer (Fig. 1b)[19,22]. Considering the center

[1]Center for Monsoon System Research, Institute of Atmospheric Physics, Chinese Academy of Sciences, 100029 Beijing, China. [2]State Key Laboratory of Numerical Modeling for Atmospheric Sciences and Geophysical Fluid Dynamics, Institute of Atmospheric Physics, Chinese Academy of Sciences, 100029 Beijing, China. [3]Laboratory for Regional Oceanography and Numerical Modeling, Qingdao National Laboratory for Marine Science and Technology, 266237 Qingdao, China. [4]Department of Atmospheric Sciences, Yunnan University, 650500 Kunming, China. ✉e-mail: huangping@mail.iap.ac.cn; wanglin@mail.iap.ac.cn

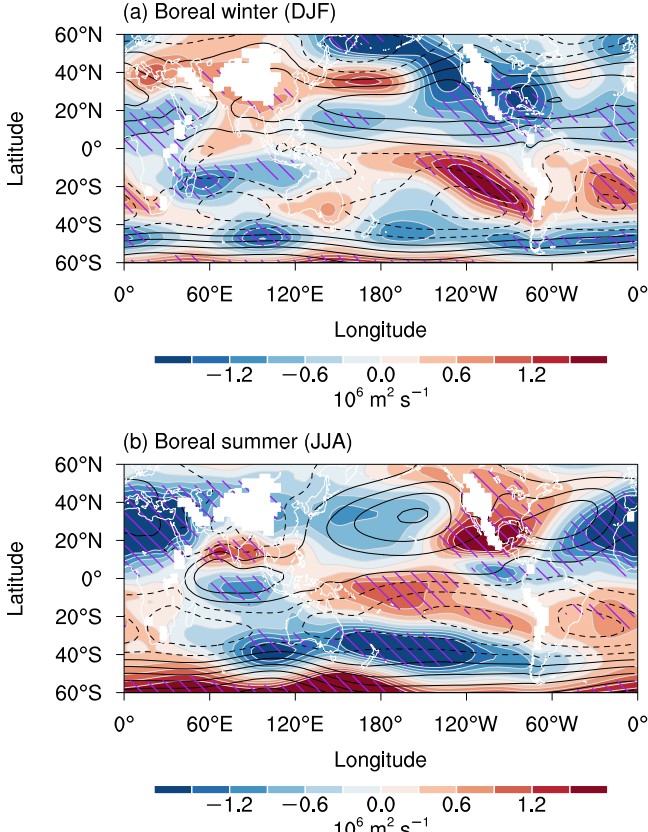

**Fig. 1 | Seasonal changes in subtropical circulation under greenhouse warming.**
**a**, **b** Changes in the 850 hPa streamfunction (shading) in the Shared Socioeconomic
Pathway 5-8.5 (SSP5-8.5) experiment relative to the historical experiment during
**a** the boreal winter (December–February; DJF) and **b** the boreal summer
(June–August; JJA). The contours in **a** and **b** represent the climatology of the
850 hPa streamfunction in the historical experiment (interval: $5 \times 10^6 \text{ m}^2 \text{ s}^{-1}$).
Hatching indicates that the change is robust (see the "Methods" section for the
details of the criteria).

and intensity of background SC seasonally vary, we cannot choose a
domain that is suitable for all seasons and distinguish the changes in
intensity and location. Due to the mixture of these factors, we did not
obtain a robust projection on global SC changes.

The uncertainty of SC changes is also associated with the multi-
plicity of the influencing factors, including the direct $CO_2$ radiative
forcing[17,30–32], sea surface temperature (SST) warming[28,30,33], and
increased atmospheric stability[34]. The direct $CO_2$ effect can induce
enhanced land–sea thermal contrast[34] with an intensification of the
Asian and northern African monsoons in boreal summer[31,32,35]. The SST
warming comprises two components: a uniform component and a
patterned component. The uniform component is closely linked to the
magnitude of global-mean surface warming, which weakens the tro-
pical overturning circulation[23,24,36] and increases low-level water vapor
with further intensification of the hydrological cycle[24,37,38]. By contrast,
the patterned SST warming can induce a shift in tropical convection
and circulation[39–41]. These mechanisms can result in specific local cir-
culation changes[27], but the delicate offset of different physical pro-
cesses differs among the climate models participating in Phase 6 and 5
of the Coupled Model Intercomparison Project (CMIP6 and
CMIP5)[19,22,28,42], leading to uncertain projections of the global SC
changes.

Here, we present a method to extract intensity change in global
SC independent of the domain and season selection. A robust
weakening of global SC is projected under the Shared Socio-
economic Pathway (SSP) 5-8.5 scenario by the end of the 21st century

in 34 CMIP6 models and in the abrupt-4×CO2 experiments forced by
an abrupt quadrupling of the pre-industrial $CO_2$ level in 32 CMIP6
models. The robustness of the weakening is verified by the single-
model initial-condition large ensemble (SMILE)[43] and the 'Database
for Policy Decision Making for Future Climate Change' (d4PDF)
simulations[44], which consider the uncertainties from internal varia-
bility and SST warming pattern, respectively. We clarify that the
robust weakening of global SC is dominated by global-mean surface
warming and partly counteracted by the direct $CO_2$ effect on dif-
ferent timescales, confirmed by the third Phase of the Cloud Feed-
back Model Intercomparison Project (CFMIP-3) in CMIP6[45]. This
unified mechanism highlights the dependence of projected SC
changes on simulation scenarios, thus providing an explanation for
prior uncertainties in projections.

## Results
### Robust weakening in the subtropical circulation
The overall intensity change of SC is defined by the projection of global
(10°–45°S, 10°–45°N) SC changes onto the SC climatology (see the
"Methods" section for details). Figure 2 shows the intensity change in
global SC in several sets of experiments, including the SSP5-8.5,
abrupt-4×CO2, CFMIP-3, SMILEs, and the d4PDF (see the "Methods"
section for details). The multiple experiments illustrate the impacts of
different mechanisms. The intensity changes in SSP5-8.5 (black dots in
Fig. 2) and abrupt-4×CO2 (orange dots in Fig. 2) show a weakened SC in
most models and both hemispheres (Supplementary Figs. S2 and S3),
although the degree shows a large inter-model spread. The inter-
model correlation coefficient between the intensity change in global
SC and global-mean surface warming among the models is −0.43
($P < 0.02$) in SSP5-8.5 and −0.55 ($P < 0.002$) in abrupt-4×CO2, sug-
gesting that the degree of weakened global SC is related to the degree
of global warming. The weakening is more apparent in the abrupt-
4×CO2 experiments (orange dots in Fig. 2) and in both hemispheres,
with a stronger global-mean surface warming than SSP5-8.5.

We also confirm the earlier conclusion of a decline in the diver-
gent SC resulting from the weakening of the tropical overturning
circulation[23–26]. As shown in Supplementary Fig. S4, the divergent
component of SC, represented by the velocity potential at 850 hPa,
demonstrates a more robust decrease in the SSP5-8.5 scenario, align-
ing with previous findings[23–26].

As the internal variability could greatly influence the regional SC
changes[27], we analyze the SMILEs of five models, CESM1-CAM5,
CanESM2, CSIRO-Mk3-6-0, GFDL-CM3, and MPI-ESM, and confirm a
limited role of internal variability to the weakened SC (gray hollow
markers in Fig. 2). An apparent exception projecting a strengthened
global SC is MPI-ESM (Fig. 2), including the 100 members of MPI-ESM in
the SMILEs, and the related versions of MPI-ESM in the SSP5-8.5 (MPI-
ESM1-2-HR and MPI-ESM1-2-LR) and abrupt-4×CO2 (MPI-ESM-1-2-HAM,
MPI-ESM1-2-HR, and MPI-ESM1-2-LR) from the CMIP6. Even though, the
results of MPI-ESM-related models follow the negative correlation
between global SC change and global-mean surface warming, reflect-
ing the enhanced SC could be related to the lower warming in the
models.

The uncertainty from patterned SST warming is also evaluated by
the simulations from d4PDF, which are six atmosphere-only experi-
ments forced by 4-K patterned warming scaled from six representative
SST warming patterns (Supplementary Fig. S5) extracted from the
CMIP5 models (purple markers in Fig. 2). The spread of the intensity
changes in the six d4PDF experiments does not exceed the inter-model
spread of SSP5-8.5 and abrupt-4×CO2 experiments, also exhibiting a
robust weakening of global SC. We contrast two extreme results in the
six d4PDF experiments: one involving slight intensification and the
other a considerable weakening (Fig. 2), particularly in the Southern
Hemisphere (Supplementary Figs. S2 and S3). These results are asso-
ciated with HFB_4K_MI (Supplementary Fig. S5d) and HFB_4K_MR

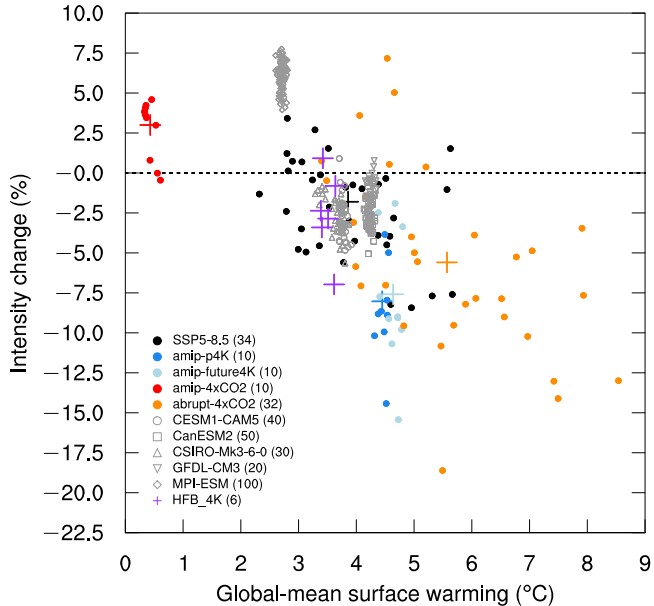

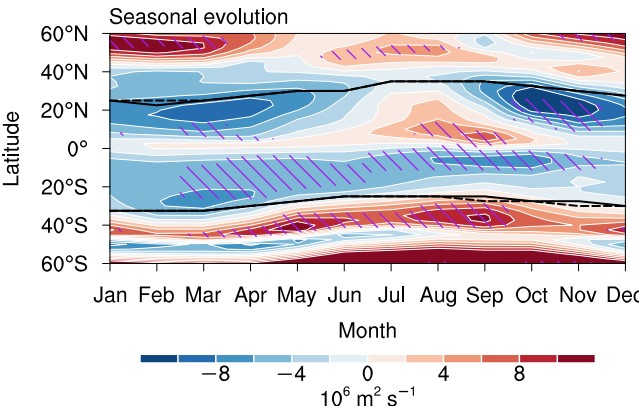

**Fig. 3 | Seasonal and latitudinal intensity changes in the 850 hPa streamfunction (shading) in the Shared Socioeconomic Pathway 5-8.5 (SSP5-8.5) experiment relative to the historical experiment.** The black solid (dashed) curves indicate the center of the subtropical circulation in each hemisphere in the historical (SSP5-8.5) experiment (see the "Methods" section for details). Hatching indicates that the change is robust (see the "Methods" section for the details of the criteria).

**Fig. 2 | Intensity changes in the global subtropical circulation with respect to global-mean surface warming in several sets of experiments.** The intensity changes in the global subtropical circulation in Shared Socioeconomic Pathway 5 (SSP5-8.5) runs (years 2070–2099; black dots; 34 models) relative to historical runs (years 1979–2008), in amip-p4K (years 1979–2008; blue dots; 10 models) relative to amip experiment (years 1979–2008), in amip-future4K (light-blue dots) relative to amip experiment, in amip-4×CO2 (red dots) relative to amip experiment, in abrupt-4×CO2 (years 121–150; orange dots) relative to piControl experiment (second 100 years), in SMILEs (single-model initial-condition large ensembles) from CESM1-CAM5 (years 2070–2099 relative to years 1979–2008; gray hollow circles; 40 ensembles), in SMILEs from CanESM2 (gray hollow squares; 50 ensembles), in SMILEs from CSIRO-Mk3-6-0 (30 ensembles), in SMILEs from GFDL-CM3 (gray hollow inverted triangles; 20 ensembles), in SMILEs from MPI-ESM (gray hollow rhombuses; 100 ensembles) and in six HFB-4K experiments (purple crosses) relative to HPB experiment from the Database for Policy Decision Making for Future Climate Change (d4PDF). For the changes in each model expressed as dots, the cross with the same color represents their multi-model mean.

(Supplementary Fig. S5f), respectively. In HFB_4K_MI, a rare cooling occurs over the Southern Ocean, enhancing the meridional temperature gradient and strengthening westerlies[46]. Additionally, there is a relatively weak El Niño-like warming in the tropical Pacific, resulting in a weaker westerly change over the equatorial western Pacific. Both these features in HFB_4K_MI contribute to strengthening the Southern Hemisphere SC.

The role of the three primary processes in $CO_2$ increase, the uniform SST warming, the patterned SST warming, and the direct $CO_2$ radiative forcing[22,30], are investigated using a set of atmosphere-only experiments from CFMIP-3[45], including amip, amip-p4K, amip-future4K, and amip-4×CO2. Uniform (blue dots in Fig. 2) and patterned (light-blue dots in Fig. 2) SST warming both significantly weaken the global SC. By contrast, the direct $CO_2$ effect strengthens the global SC confirmed by the amip-4×CO2 experiment (red dots in Fig. 2), opposite to the effect of SST warming. On a global scale, the diminishing influence of SST warming overcomes the enhancement of the direct $CO_2$ effect, leading to a robust weakening, although these two mechanisms are comparable in some regions[22].

The effects of SST warming and direct $CO_2$ radiative forcing have distinct temporal features. The direct $CO_2$ effect is a rapid process with timescales from weeks to months[22,34], whereas the SST warming associated with its impact is almost proportional to the relatively slow global-mean surface warming. The opposite roles of SST warming and direct $CO_2$ effect with distinct temporal features can explain the more robust decrease in abrupt-4×CO2 than in SSP5-8.5 experiments (Fig. 2).

There is persistent new $CO_2$ emission in SSP5-8.5[47] but not in abrupt-4×CO2, resulting in more transient response in SSP5-8.5 during the years 2070–2099 than in abrupt-4×CO2 during the years 121–150. Given that the warming of the transient response to a $CO_2$ emission is weaker compared to the response nearing equilibrium, the role of uniform SST warming is underrated in SSP5-8.5 relative to the response nearing equilibrium in abrupt-4×CO2. This process could be the reason for the absence of a robust change obtained in many previous studies utilizing SSP5-8.5 experiments.

## Seasonal variation of the changes in the subtropical circulation

As SC and the associated processes are seasonally and latitudinally varying[1], we further investigate the seasonal and latitudinal SC change by projecting the SC onto the climatological SC latitude-by-latitude and month-by-month (see the "Methods" section for details). Figure 3 shows the seasonal and latitudinal intensity changes in SC of the multi-model mean of CMIP6 models. The center of the climatological SC is shown as a reference for the SC changes (see the "Methods" section for details; Supplementary Fig. S6). The Northern Hemisphere SC is weakened around the equatorward flank of the climatological SC center throughout the year, except in JJA (Fig. 3). The Southern Hemisphere SC is weakened mainly on the equatorward flank of the climatological center but strengthened on the poleward flank (Fig. 3), resembling a poleward shift.

The weakened SC is not robust in all months and latitudes (see the definition of robustness in the "Methods" section), implying a more intricate mechanism underlying these changes in SC. In the amip-p4K simulation, uniform SST warming results in a robust weakening of global SC throughout the year (Fig. 4a and Supplementary Fig. S7). This weakening shifts seasonally with the climatological SC center and occurs primarily on its equatorward flank in a few months, possibly linked to the more pronounced circulation weakening in the tropics than in the extratropics[23]. The impact of the patterned SST warming (Supplementary Fig. S8) on the SC with the impact from uniform SST warming removed (Fig. 4b) is relatively weak compared to the uniform SST warming (Fig. 4a), but it enhances the poleward flank of the SC in the Southern Hemisphere during the austral summer (Fig. 4b and Supplementary Fig. S9). This enhancement is associated with the southeasterly wind changes over the southeastern Pacific (Supplementary Fig. S8), which is modulated by a stronger west-minus-east gradient during the austral summer around 20°S–40°S of the prescribed-SST change pattern (Supplementary Fig. S8)[48]. The results in the d4PDF

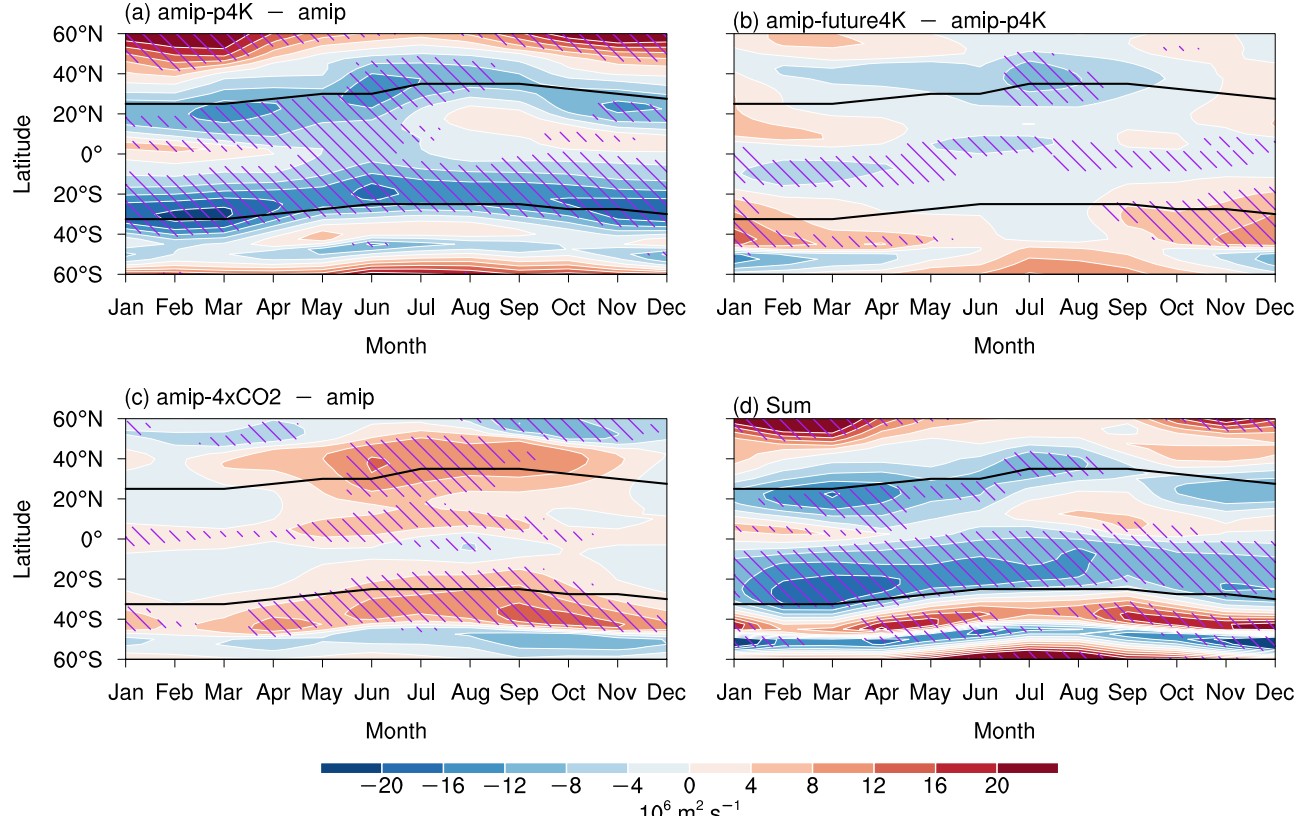

**Fig. 4 | Seasonal and latitudinal intensity changes in the global subtropical circulation in the atmosphere-only experiments. a–c** Seasonal and latitudinal intensity changes in 850 hPa streamfunction under **a** uniform sea surface temperature (SST) warming, **b** patterned SST warming, and **c** direct $CO_2$ radiative forcing. The black curves indicate the center of the subtropical circulation in each hemisphere. **d** Sum of (**a–c**). Hatching indicates that the change is robust (see the "Methods" section for the details of the criteria).

simulations with six different SST change patterns (Supplementary Fig. S10) are consistent with those in SSP5-8.5 (Fig. 3), also suggesting a weak influence of the uncertainty in the SST change pattern.

By contrast, direct $CO_2$ radiative forcing strengthens the global SC, especially during the boreal summer monsoon season (June–September; Fig. 4c and Supplementary Fig. S11). Strengthening of the SC is stronger on the poleward flank of the climatological SC center because of the superposition of the poleward shift in the mid-latitude jet (Supplementary Fig. S12). Direct $CO_2$ radiative forcing can induce stratospheric cooling and a poleward shift of midlatitude jet throughout the year[46,49], favoring the strengthening of the SC in both hemispheres. However, its impact is located around the midlatitude jets, exactly the poleward boundary of the climatological SC in both hemispheres (Fig. 4c and Supplementary Fig. S12), indicating the poleward shift in the midlatitude jet could not be the major factor enhancing SC, especially during the boreal summer.

The combined impact of the uniform and patterned SST warming, and direct $CO_2$ radiative forcing (Fig. 4d) closely resembles the SC changes in the SSP5-8.5 runs (Fig. 3). Some minor discrepancies between Figs. 3 and 4d still appear in Northern Hemisphere during the boreal autumn, implying a relatively weak role of the atmosphere-ocean coupled process that is neglected in the atmosphere-only experiments. Compared with the SC changes in the SSP5-8.5 runs (Fig. 3), the weakening of SC is more robust in most months and latitudes in the Southern Hemisphere (Fig. 4d), which should be associated with the stronger warming in the +4-K amip experiments than the SSP5-8.5 runs (Fig. 2).

The weakened SC under uniform SST warming is robust throughout the subtropics (Fig. 5a), whereas the SC is strengthened in June–September under direct $CO_2$ radiative forcing (Fig. 5b), except

over the Atlantic Ocean. The low-level SC weakening under uniform SST warming is spatially homogeneous throughout the subtropics, which is consistent with the decreased vertical velocity in the mid-level troposphere (Fig. 5c)[23] and reflects the mechanism of the Sverdrup balance connected with the weakened tropical circulation[1]. The degree of weakening in low-level SC is weaker than that in mid-level circulation and is close to that in the Hadley cell[50], given that the low-level SC is closely related to the descending branch of the Hadley cell. The major contribution to the slowdown in the tropical overturning circulation is from the Walker circulation, but not the Hadley cell[23,24,50]. The direct $CO_2$ effect mainly strengthens the SC in the boreal summer monsoon season, most apparent in the low level, by enhancing the land–sea thermal contrast[22].

The land–sea thermal contrast (Supplementary Fig. S13) could induce tropical diabatic heating and then influence the SC in both hemispheres through stationary barotropic Rossby waves[33,51,52]. The process is verified by a linear baroclinic model (LBM; see the "Methods" section for details). According to the boreal summer changes in rainfall under direct $CO_2$ effect (Fig. 5d), we select six regions with idealized heating and cooling to force the LBM in the respective experiment (Supplementary Figs. S14 and S15). The steady responses to the heating and cooling show that the diabatic heating in North Africa and cooling in the northeastern Pacific contribute most to the strengthened SC (Fig. 5b and Supplementary Fig. S16). The tropical responses are close to a Matsuno–Gill pattern[53] dependent on the location of forcing.

### Timescales of responses

The distinct temporal features of SST warming and direct $CO_2$ effect cause their relative contributions to vary across different emission

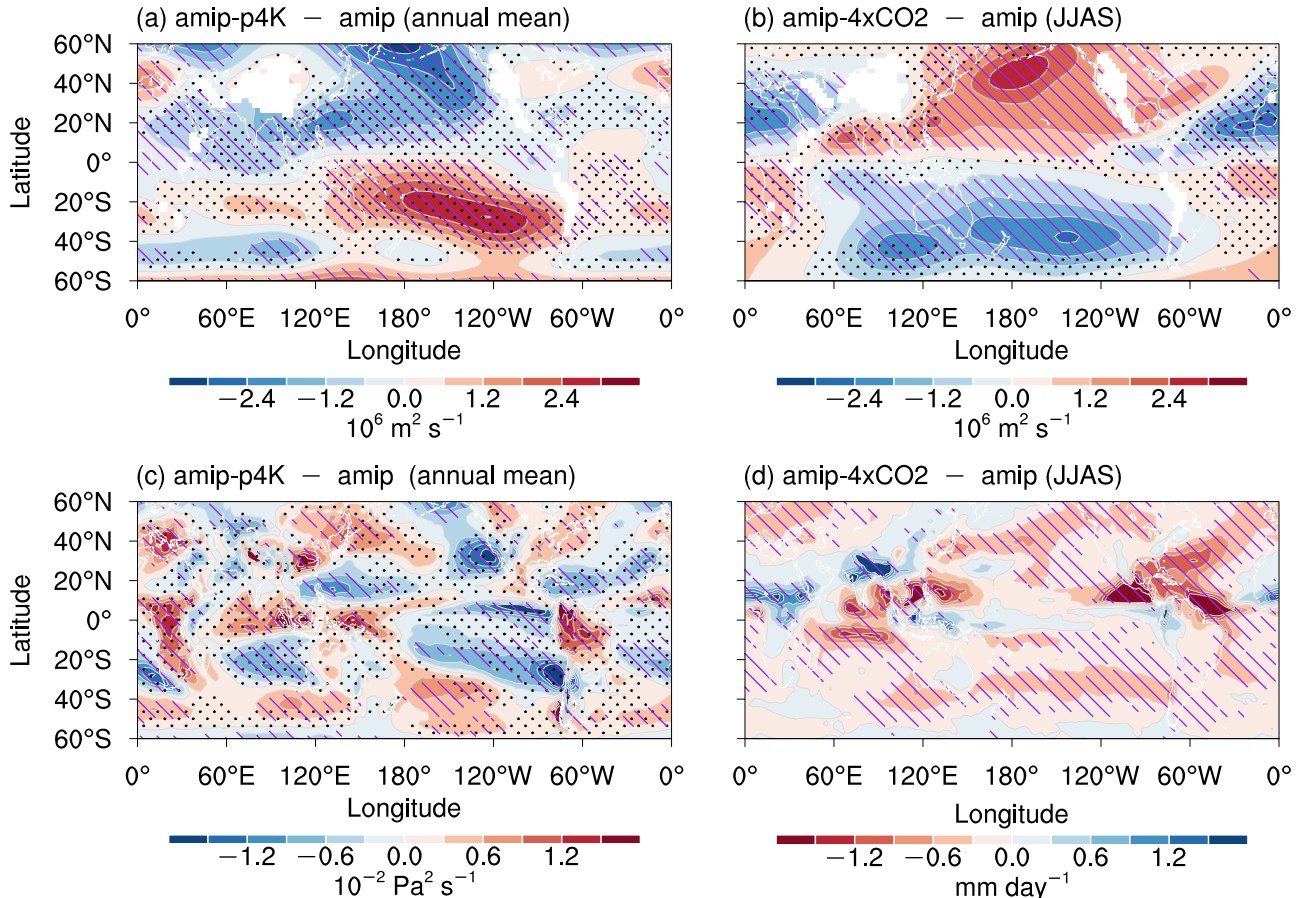

**Fig. 5 | Changes in the low-level subtropical and the mid-level vertical circulation in the atmosphere-only experiments. a**, **c** Annual changes in **a** 850 hPa streamfunction and **c** 500 hPa vertical pressure velocity under uniform sea surface temperature (SST) warming. **b** The same as **a** but for the boreal summer (June–September; JJAS) changes under direct CO$_2$ radiative forcing. **d** Boreal summer changes in rainfall under direct CO$_2$ radiative forcing. Hatching indicates that the change is robust (see the "Methods" section for the details of the criteria). Stippling in **a**–**c** indicates that the changes and their corresponding climatologies are opposite in sign, which is equivalent to local weakening.

scenarios. This property is used to verify their combined role by explaining the time-varying SC in the abrupt-4×CO2 simulation. In abrupt-4×CO2 (Fig. 6a), the direct CO$_2$ effect—the enhanced land–sea thermal contrast—reaches its peak in the first year strengthening SC. Then the weakening of SC is increasingly robust in the low-level SC, along with the increase of slow SST warming overcoming the direct CO$_2$ effect. The rate of weakening in the low-level SC at the end of the abrupt-4×CO2 experiment is about −1.14% (with a 5–95% range of −2.90% to 0.21%), per 1 K global surface warming.

The enhanced SC by the direct CO$_2$ effect appears in all latitudes and seasons in the first year of the abrupt-4×CO2 experiment (Fig. 6b). The Northern Hemisphere SC changes are slightly shifted from the boreal summer to spring and early summer. One possible reason is that the change in a single year is susceptible to internal variability, while another possibility is that the annual global-mean surface warming is around 1 K in the first year, indicating that the influence of SST warming has already occurred. However, the response in the Southern Hemisphere (Fig. 6b) closely resembles that in the atmosphere-only model (Fig. 4c), confirming the direct CO$_2$ effect on the SC in the coupled model. As greenhouse warming continues, the SC changes during years 21–30 in the abrupt-4×CO2 experiment (Fig. 6c) resemble those at the end of the SSP5-8.5 scenario (Fig. 3). In years 141–150 of the abrupt-4×CO2 experiment, the Southern Hemisphere SC is clearly weakened throughout the year, and the Northern Hemisphere SC is also weakened, except in August–September (Fig. 6d). Overall, the combined roles of SST warming and direct CO$_2$ radiative forcing in the

low-level SC changes shown here explain well the time-varying SC changes in the abrupt-4×CO2 simulations.

## Discussion

The present study reveals a robust weakening in global lower-tropospheric SC under greenhouse warming, based on a metric projecting the SC changes onto the SC climatology. The weakening of global SC is robust in the SSP5-8.5 and abrupt-4×CO2 simulations participating in CMIP6. The SC weakening is increasingly evident with the rise in global-mean surface temperature. The uncertainties from internal variability and future SST warming pattern do not apparently influence this conclusion, verified by the SMILE and d4PDF simulations. The robust weakening of SC is mainly induced by global-mean surface warming and partly counteracted by the direct CO$_2$ effect. Uniform SST warming tends to weaken the global SC throughout the year, mainly on the equatorward flank of the climatological SC center, whereas direct CO$_2$ forcing strengthens the global SC on the poleward flank during the boreal summer monsoon season. The combined mechanisms in which the effect of SST warming and direct CO$_2$ radiative forcing have different temporal features explain well the time-varying SC changes in the abrupt-4×CO2 experiment. The global SC weakening finally emerges in most models, when the slow effect of SST warming overcomes the fast effect of direct CO$_2$ radiative forcing.

The decrease in global SC is associated with a weakening global monsoon circulation under CO$_2$-induced global warming, suggesting a contracted monsoon range and drying at its boundary with arid

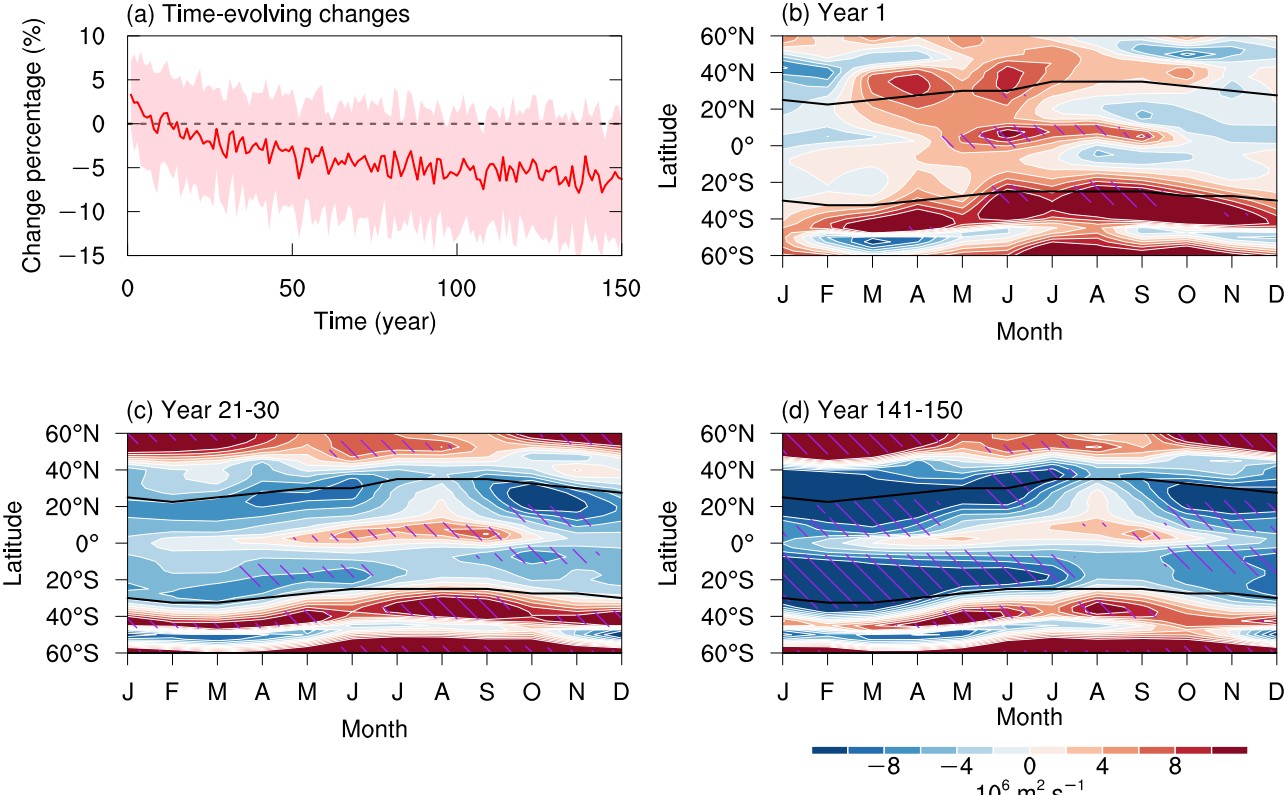

**Fig. 6 | Time evolution of the changes in the global subtropical circulation.**
**a** Multi-model mean of the intensity change in the 850 hPa streamfunction
(10°–45°S and 10°–45°N) and one inter-model standard deviation range (shading)
in the abrupt-4×CO2 experiment relative to the piControl experiment.
**b**–**d** Seasonal and latitudinal intensity changes in the 850 hPa streamfunction
during **b** year 1, **c** years 21–30, and **d** years 141–150 in the abrupt-4×CO2 experiment
relative to the climatology of second 100 years in piControl experiment. The black
curves in **b**–**d** indicate the center of the subtropical circulation in each hemisphere.
Hatching indicates that the change is robust (see the "Methods" section for the
details of the criteria).

regions, especially in the Northern Hemisphere. This implication see-
mingly contradicts the prior conclusion of a broader range of mon-
soon rainfall linked to increased moisture due to global warming[54,55].
The contradiction arises from the linkage of SC change to $CO_2$ emis-
sion trajectory and projection term, owing to the varied impact of
processes across timescales. This matter is crucial for projecting cli-
mate change under undetermined $CO_2$ emission trajectories toward
carbon neutrality[56–59]. While various CMIP6 models consistently pro-
ject a significant decline in SC, the extent of this decline varies among
models, possibly resulting from factors of uncertainty such as sensi-
tivity to $CO_2$ concentration, circulation response to uniform warming,
and land–sea contrast strength.

Although the global and seasonal variation in SC changes are
evident, significant regional disparities still exist[27]. In addition to the
intensity changes, shifts in the location of certain regional SC systems
have also been identified[19,22], which receives limited discussion in this
study. Some shifts occur in regions where multiple subsystems of SC
are interconnected, making them difficult to interpret solely in terms
of overall intensity change. Despite this, our method can efficiently
isolate overall intensity changes in global SC or in a relatively large
domain, aiding the focus on more specific changes in regional SC in
future research. Furthermore, once the domain selection is estab-
lished, this method can be employed to analyze regional SC changes
for distinguishing the changes in intensity and in location.

## Methods
### CMIP6 simulations
The historical simulation and the SSP5-8.5 experiment from 34 CMIP6
models[60] are used to represent the present-day and future climates,
respectively. The simulation skills of historical SC in these models are

evaluated by comparing them with the European Centre for Medium-
Range Weather Forecasts Reanalysis v5 (ERA5) reanalysis dataset[61]. As
shown in Supplementary Fig. S17, the Taylor diagram[62] for the spatial
correlation coefficients and standard deviations of the 850 hPa
streamfunction from 60°S to 60°N illustrates that most models show a
good performance on the simulation of SC. Thus, we do not further
pick out models in the present analyses. The pre-industrial control
simulation (piControl) and the simulation forced by an abrupt quad-
rupling of the pre-industrial $CO_2$ level (abrupt-4×CO2) from 32 CMIP6
models are also used to study the timescales of different processes.

A set of atmosphere-only experiments[45], including the amip,
amip-p4K, amip-4×CO2, and amip-future4K from 10 CMIP6 models,
are used to investigate the role of different processes. The amip
experiment is a control run forced by the observed monthly mean SST
and sea ice concentration, whereas the amip-p4K, amip-4×CO2, and
amip-future4K experiments are further forced by uniform SST warm-
ing, quadrupling of the pre-industrial $CO_2$ level and patterned SST
warming derived from the CMIP3 coupled models, respectively. The
prescribed SST warming pattern for each month is shown in Supple-
mentary Fig. S8. The present study employs the 30 years, 1979–2008,
of all the amip experiments. The details for selected models in the
different experiments are shown in Supplementary Table S1.

We only select the first simulation of the ensembles for each
model. All the monthly model data are interpolated onto a 2.5° × 2.5°
grid (90°S–90°N, 0°–357.5°E) before analysis. This study presents the
results as the multi-model mean unless stated otherwise.

### Single-model initial-condition large ensembles
To evaluate the role of internal variability, five single-model initial-
condition large ensembles (SMILEs) are used in this study[43], which are

CESM1-CAM5, CanESM2, CSIRO-Mk3-6-0, GFDL-CM3, and MPI-ESM with 40, 50, 30, 20, and 100 members, respectively. The historical and representative concentration pathway 8.5 (RCP8.5) simulations are used.

## Database for policy decision making for future climate change

The large ensemble simulation named 'Database for Policy Decision Making for Future Climate Change' (d4PDF) is used to study the role of uncertainty in future SST warming patterns[44]. The Meteorological Research Institute AGCM, version 3.2 (MRI-AGCM3.2) used here includes the simulations for a present climate (1951–2010) with 100 members (HPB) and a future climate for 60-year integration with 90 members. In this model, the global-mean surface air temperature of the future climate is about 3.6 K warmer than that of the present climate, corresponding to the condition around the end of the 21st century under the RCP8.5 scenario[44]. These simulations consist of six experiments, each using a distinct SST change pattern from CMIP5 models (HFB_4K_CC, HFB_4K_GF, HFB_4K_HA, HFB_4K_MI, HFB_4K_MP, HFB_4K_MR; Supplementary Fig. S5) to capture the impacts of SST warming pattern diversity. For each experiment, 15 members for a 60-year future warming climate are conducted using different initial conditions and different small perturbations of SST. The first 15 ensemble members from 1979 to 2008 in historical simulation are used to represent the historical climate in this study. Different from the future patterned-SST forced experiments such as amip-p4K and amip-future4K from the CFMIP-3 in CMIP6, the greenhouse gases are set to the same value as the end of the RCP8.5 scenario in the global warming simulation from the d4PDF[44]. Thus, the intensity changes of global SC in the 4-K future climate simulations in d4PDF are lower than that in the amip-p4K and amip-future4K experiments and are close to the results in the SSP5-8.5 scenario.

## Definition of the SC

In this study, we focus on the dominant rotational component of SC at the low level[63], represented by the 850 hPa streamfunction. The meridional center of SC for each month is defined as the maximum of the climatological zonal-mean 850 hPa streamfunction in the Northern Hemisphere (20°–40°N) and the minimum of the climatological zonal-mean 850 hPa streamfunction in the Southern Hemisphere (20°–40°S) in the historical, amip and piControl experiments. The SC center and the climatological 850 hPa streamfunction in the historical experiment are shown in Supplementary Fig. S6.

## Intensity change

The overall intensity change in global SC is defined by the projection of the changes in global SC (10°–45°S and 10°–45°N) onto its 12-month climatology calculated as:

$$A = \frac{\sum_{k=1}^{n}\sum_{j=1}^{n}\sum_{i=1}^{n}\left(X_{ijk}Y_{ijk}\right)}{\sum_{k=1}^{n}\sum_{j=1}^{n}\sum_{i=1}^{n}\left(Y_{ijk}Y_{ijk}\right)} \qquad (1)$$

where $A$ represents the intensity change (unit: %); $X$ and $Y$ represent the change and climatology of a circulation system, respectively; $i$, $j$, and $k$ represent the latitude, longitude, and month dimensions, respectively. This definition is independent of the domain and the seasonal evolution of SC. We also test other latitudinal ranges, including 45°S–45°N, 10°–40°S, and 10°–40°N and 10°–50°S and 10°–50°N, which do not influence the conclusions (Supplementary Fig. S18). The projection method defining the percentage intensity change prevents unreasonable large values in cases where the climatological SC is close to 0, unlike the conventional method that divides changes by climatology.

Considering SC is seasonally and latitudinally varying, we also project the SC change on the climatological SC latitude-by-latitude and

month-by-month calculated as:

$$A_{ik} = \frac{\sum_{j=1}^{n} X_{ijk}Y_{ijk}}{\sqrt{\sum_{j=1}^{n} Y_{ijk}Y_{ijk}}} \qquad (2)$$

## Test of robustness

A change is considered robust if it meets two criteria: the ratio of multi-model mean to inter-model standard deviation of the change is >1 and at least 66% of the models show a change greater than the internal-variability threshold $\gamma$. This test of robustness considers both the uncertainty across the models and internal variability, which is similar to the approach used in the IPCC AR6 WG1 report[64]. The ratio test is stricter than the sign agreement test often used in previous studies to evaluate the inter-model robustness. The internal-variability threshold is defined as $\gamma = \sqrt{2} \cdot 1.645 \cdot \sigma_{30yr}$, where $\sigma_{30yr}$ is the standard deviation of 30-year climatology of thirteen periods separated from 500-year simulation in the piControl in which the first 100 years are ignored[64].

## Linear baroclinic model

We use a simple dry model, Linear baroclinic model (LBM)[65], to investigate the atmospheric response to prescribed heating[66]. The LBM applied here consists of primitive equations linearized concerning a climatological state of June–September, with a horizontal resolution of T42 (roughly equivalent to 2.8°) and 20 vertical sigma levels. The spatial patterns of prescribed heating and cooling maximum at 0.45 level are shown in Supplementary Fig. S14. The vertical profile of diabatic heating forcing the LBM is shown in Supplementary Fig. S15, with a maximum (minimum) at sigma = 0.45 level. The LBM is run up to 30 days, and the 850 hPa streamfunction in the 21–30 days is used to represent the steady response shown in Supplementary Fig. S16.

## Data availability

The CMIP6 data is available at https://esgf-node.llnl.gov/projects/esgf-llnl/. The SMILEs data is available at https://www.cesm.ucar.edu/projects/community-projects/MMLEA/. The ERA5 reanalysis dataset is available at https://www.ecmwf.int/en/forecasts/datasets/reanalysis-datasets/era5. The d4PDF is available at http://search.diasjp.net/search?lang=en&k=d4PDF.

## Code availability

The data in this study are analyzed with NCAR Command Language. The code used in this study is available on request from the corresponding author.

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

## Acknowledgements

This work was supported by the National Key Research and Development Program of China (2019YFA0606703), the National Natural Science Foundation of China (42105027 and 41975116), the China Postdoctoral Science Foundation (BX20200329 and 2020M680646) and the Youth Innovation Promotion Association of CAS (Y202025). This work was also supported by the National Key Scientific and Technological Infrastructure project "Earth System Numerical Simulation Facility" (EarthLab). We acknowledge the World Climate Research Programme's Working Group on Coupled Modeling, which is responsible for CMIP6, and the climate modeling groups for producing and making available their model output.

## Author contributions

S.Z. and P. Huang conceived the study, performed the analyses, and wrote the paper. L.W., K.H., and G.H. helped with interpretation and feedback. P. Hu helped run the LBM experiments. All authors approved the final manuscript.

## Competing interests

The authors declare no competing interests.
