## [Peer Review File · Nature Communications]

Robust changes in global subtropical circulation under greenhouse warmingReviewers' comments:

Reviewer #1 (Remarks to the Author):

"Exploring Robust Changes in Global Subtropical Circulation Under Greenhouse Warming"

This article investigates the changes occurring in the subtropical circulation (SC) in the lower troposphere as a result of a warmer climate. By analyzing CMIP6 models, the study reveals significant global SC changes characterized by a weakening trend in the Northern Hemisphere throughout the year, except during the boreal summer, and a poleward shift in the Southern Hemisphere. The authors attribute these changes primarily to global-mean warming, which weakens the SC, particularly in its equatorward flanks. However, they also note that direct CO₂ radiative forcing strengthens the SC in the poleward flanks, specifically during the boreal summer monsoon season. While the article focuses on an intriguing topic, further clarification is necessary before publication.

There is a need for more detailed explanations regarding the role of changes in local sea surface temperature (SST) compared to tropical SST and their varying impacts. The study should delve into the seasonal variability of the crucial role played by tropical diabatic heating in SC.

Furthermore, the changes in SC are not zonally symmetric in both hemispheres. For instance, during boreal winter, the SC over the South Pacific exhibits greater strength compared to the South Atlantic or Indian regions. The SC over the South Pacific is known to be influenced by the heating over SPCZ (Fahad et al. 2021). The article would benefit from an extended discussion focusing on these discrepancies, with particular emphasis on the local forcing and remote influences.

The article would also benefit from a more comprehensive discussion, specifically in the section titled "Role of SST warming and direct CO₂ radiative forcing," which should concentrate on individual hemispheres and basins.

There is a need for clarification on the statistical significance of the findings mentioned in lines 173-175.

In the summary section, it is crucial to incorporate a discussion on how the results of this article differ or align with existing literature, particularly in cases where similar research has been previously conducted, particularly in the Southern Hemisphere.

Reviewer #2 (Remarks to the Author):

This study examines the changes in global subtropical circulation (SC) under greenhouse warming using the future projections from CMIP6 models. The results suggest a weakening in the Northern Hemisphere

SC and a poleward shift in the Southern Hemisphere SC. The authors further discuss the contributions of SST warming and direct CO₂ radiative forcing to the projected changes. Although this study provides some insights into future SC changes and their possible mechanisms, there are a few major flaws in the current version, making this study not appropriate to be published in Nature Communications. I would encourage the authors to resubmit the manuscript to other journals which accept longer articles. The detailed comments are listed below.

Major comments:

1. "Robust changes". The authors have never defined how they determine "robust" in the SC changes. Is it based on a large model agreement, the significance of the change, the seasonal consistency, or any others? This is a piece of important information that the authors would need to include.
2. Abstract. Overall, the abstract is too technical. Please rewrite it in a form that is more understandable to the public audience.
3. Introduction. The introduction is very unclear about what is still unknown in the SC changes, and what is new in this study compared to the previous literature. The introduction also lacks examples of how SC changes may impact the weather and climate. The motivation is unclear. Although the authors have cited some references, their discussion of these previous studies is not thorough enough. It is important to include a discussion in terms of the significance of this study.
4. Novelty. There have been a great number of studies using CMIP6 models to study SC changes, so what is the novelty of this study?
5. Method. This section does not provide enough information. For example, how the 34 models were selected. Are they based on the data availability or the model fidelity in simulating the SC? The baseline is the model would need to capture the SC well, as well as the ensemble members the authors used. Please discuss how many ensemble members are used in each model and what ensemble members the authors used. In addition, I am not sure why a short period is examined in this study. The historical run is up to 2014 while the authors only used the runs up to 2000. I would encourage the authors to extend their period to reduce the uncertainty and have more robust results. Please also clarify why the historical runs are used from 1971. Most of the reanalysis data are from 1979. It is better to match the period between historical runs and reanalysis data when comparing the model fidelity in simulating the SC which is currently missing. I understand the authors want to use the multi-model mean for an easier interpretation of the results, it is still with significant scientific meaning that the authors would need to discuss the model spread. Especially as pointed out by the authors that there are large inconsistencies between the studies which imply a large model spread. This is one of the novelties that I can see from this study. In terms of the definition of the subtropical circulation, it is not sure why the center of subtropical circulation is calculated using a large latitude spread which includes the tropics. The authors would need to test the sensitivity of their results by adjusting this latitude range.
6. Readability. The authors want to make the manuscript concise, however, they lost clarity. Most of their description or statements are too general and hard to follow. Their figures are generally not straightforward, so it will be much more helpful if they could elaborate more in the main text. The mechanisms they discuss are also not clear and missing details. They would need to refine most of their main text to keep the manuscript concise while also making the statement clear and with enough details for the readers to understand.
7. Figures. The stipplings are very hard to see in the figures. For Figure 2, please do not make -4 to 4 white. There is a large model agreement in the tropics but there is no color. Figure 3: The hatched

pattern makes the figure messy and the stippling is rather hard to see. I think the authors could think of a better way to show the local weakening. Figure 4d: how this figure is calculated would need to be discussed in the Method section. And why 40S-40N is used by including the tropics?

8. Discussion. A comprehensive discussion is necessary following the summary in terms of consistency with the previous studies and the implication we can get from this paper.

9. Please provide a map showing the SST warming pattern in the supplementary. Or provide a more detailed description of the amip experiments such as the value of the warming, and the pattern of the warming.

10. Because the introduction is poorly written, it is hard to determine the significance and novelty of this manuscript. Please rewrite the introduction to show the novelty of this study. In addition, the manuscript is poorly written as I mentioned that some important details are missing, also the figures are hard to understand. Please try to include as many details in the Method section in terms of calculation and analysis methods. Most of the discussion cited mechanisms from previous papers and the main findings seemed to be the role of SST warming and CO2 forcing. Therefore, this paper is more of a further analysis instead of a breakthrough finding. I would therefore recommend transferring this study to another journal. I would also encourage the authors to try to convert the manuscript into a longer paper to include more details and thorough discussion.

Minor comments:

1. L6-8. L13-16. L18-20. Please rephrase, hard to understand.

2. L21-22. How do the results carry implications for subtropical climate changes?

3. L33-34. Is this something already found or the authors' hypothesis? Please clarify.

4. L33. "but no consistent response..." For example?

5. L34-37. Not linked to the previous sentence as to which mechanisms are in which seasons and which regions?

6. L37-38. It seems that it has been a well-established mechanism from the impact of the direct CO2 effect. And this mechanism has been used to explain changes in SC during summer. So, what is new in this study?

7. L41. The mechanism is fairly unclear here.

8. L43. Same comment as above.

9. L48. The authors have never explained why there is a need to examine the changes latitude-by-latitude.

10. L50-52. Please do not repeat this again. This has been discussed in the abstract.

11. L53-57. Please do not include a summary of the results here. The introduction should be more focused on the background of the field and the significance and novelty of this study.

12. Need some description of what the SC changes are like and how they change in the figure when discussing Fig. 1.

13. L65-67. How the changes in the tropical circulation contribute to the changes in SC. The authors need to clarify that.

14. L65. I do not see the tropical circulation change as seasonally consistent from Fig. 1a and 1b. They need better quantification.

15. L68-70. This kind of discussion is good. The authors would need to have some similar discussion to indicate how the SC changes and its implication to the changes of the climate systems.

16. L71. "and latitude". I do not see this from the earlier discussion.

17. L77-78. The SC in the winter season is more of an indication of a weakening in the southward flank rather than a weakening of the center when comparing the center of SC and the center of SC changes.
18. L85. Briefly describe
19. L90-91. Briefly describe their differences.
20. L99-100. Why this is specifically discussed? Unless this is important to some weather and climate systems.
21. L101. Not always, only in a few months.
22. L104. Please elaborate. The authors always make a general statement. They could state in more detail such as which latitudes and which seasons.
23. L106-107. Not a convincing mechanism here. The reduction in midlatitude baroclinicity is dominant in which seasons and caused by which SST change patterns? Please provide more details.
24. L138-139. Can this mechanism explain why the strengthening most occurred in the poleward flank?
25. L355. Not straightforward how the seasonal changes are calculated. Needed a brief clarification here or in the main text.
26. L385. Where is the shading?

Reviewer #3 (Remarks to the Author):

This study investigates changes in the global subtropical circulation in response to greenhouse warming and tries to quantify the effects of the global-mean warming and the direct CO₂ radiative forcing by analyzing the amip, amip-p4k, amip-future4K and amip-4xCO₂ experiments in CMIP6. Authors highlight that combining these two mechanisms results in seasonally and latitudinally dependent changes in the subtropical circulation under a warmer climate. This study provides systematic analyses with different amip-type experiments in CMIP6. However, the main results are consistent with previous studies and not novel enough for Nature Communication. Thus, I cannot recommend the manuscript for publication in nature Communications.

Major comments:

1. Significance: Many studies have already discussed the roles of global-mean warming, the patterns of SST changes, and direct CO₂ radiative forcing on the changes in the subtropical circulation as referred to in the manuscript. The main results of this study, such as the weakening of tropical circulation, the intensification of the Asian summer monsoon, and the poleward shift of SH subtropical circulation, confirm results from many previous studies. Thus, it seems that the main results are not novel and significant enough for Nature Communication.
2. Validity/Data and Methodology: The description of the data and methodology is well provided. However, I think the comparison of AMIP-future4K and AIMP-p4K is not enough to address the role of the SST pattern changes. One of the dominant uncertainties in circulation changes is rooted in uncertainties in the future SST pattern changes (e.g., Chadwick et al., 2014 referred to in the

manuscript). Since the AMIP-future4K uses the same mean SST pattern projected by CMIP simulations, the current analysis has limitations in addressing the effect of future SST pattern changes. However, models tend to show robust responses to global-mean warming and direct CO₂ radiative forcing as discussed in many previous studies (e.g., Fahad et al., 2021; Bony et al., 2013; Vecchi et al., 2006; and many others referred to in this manuscript).

3. Clarity and Context: The manuscript is well written and clear. However, there are several limitations in context. First, the main results are only based on mean of multi-model changes. Fidelity of model simulation in the present and the role of model uncertainties in addressing future changes should be discussed. Second, the significant test is only based on the degree of the models agree on the sign of the multi-model mean in the study. Considering large internal variability in circulation changes, a significant test should be also applied with respect to internal variability.

Response to reviewer #1:

Thank you for the insightful comments and detailed instruction on how to improve the manuscript. The quality of the manuscript has been greatly improved based on your comments. Our point-by-point reply follows, where the original comments are quoted in italic.

1. There is a need for more detailed explanations regarding the role of changes in local sea surface temperature (SST) compared to tropical SST and their varying impacts. The study should delve into the seasonal variability of the crucial role played by tropical diabatic heating in SC.

Re: Thank you for the constructive suggestion.

In our original manuscript, the analyses based on amip-p4K and amip-future4K simulations (new Fig. 4, Supplementary Figs. S7 and S8) exhibit that the impact of prescribed-SST warming pattern is relatively weak compared to the uniform SST warming on SC. Thus, we did not discuss the role of SST in different locations in the original manuscript. In the revision, we improve the related statements shown in Lines 155–162.

To further illustrate the role of SST warming pattern, we analyzed the “Database for Policy Decision Making for Future Climate Change” (d4PDF) simulations in the revision. The d4PDF has six experiments using six typical SST change patterns in CMIP5 models respectively (HFB_4K_CC, HFB_4K_GF, HFB_4K_HA, HFB_4K_MI, HFB_4K_MP, HFB_4K_MR; Supplementary Fig. 4) to produce the impacts of the diversity in SST warming patterns. Results in the d4PDF simulations (Supplementary Fig. 9) align with those from SSP5-8.5 (Fig. 4a), indicating a limited influence of SST change pattern uncertainty. We discuss this in Lines 160–162.

In response to your suggestion, we employ a linear baroclinic model (LBM) to explore the contribution of tropical diabatic heating changes across different regions to the enhanced SC due to direct CO₂ effects. The LBM employs linearized primitive equations based on a June–September climatological state, with T42 horizontal resolution (approximately 2.8°) and 20 vertical sigma levels. A detailed LBM introduction is presented in Lines 352–361.

The LBM uses prescribed diabatic heating and cooling locations, detailed in Supplementary Fig. S13. The model runs for 30 days, with the 850 hPa streamfunction during days 21–30 representing the steady response (Supplementary Fig. 15). Steady

responses reveal that diabatic heating in North Africa and cooling in the Northeastern Pacific contribute predominantly to the strengthened SC (Fig. 5b and Supplementary Fig. 15). These tropical responses align closely with a Matsuno-Gill pattern, contingent on the forcing's location. The discussion of these results is presented in Lines 198–205.

Figure S13. The prescribed diabatic heating and cooling for six regions used in the linear baroclinic model.

Figure S15. The steady response in 850 hPa streamfunction to the prescribed diabatic heating and cooling shown in Fig. S13.

2. Furthermore, the changes in SC are not zonally symmetric in both hemispheres. For instance, during boreal winter, the SC over the South Pacific exhibits greater strength compared to the South Atlantic or Indian regions. The SC over the South Pacific is known to be influenced by the heating over SPCZ (Fahad et al. 2021). The article would benefit from an extended discussion focusing on these discrepancies, with particular emphasis on the local forcing and remote influences.

The article would also benefit from a more comprehensive discussion, specifically in the section titled "Role of SST warming and direct CO₂ radiative forcing," which should concentrate on individual hemispheres and basins.

Re: Thank you for your comment. Indeed, SC changes exhibit hemispheric asymmetry, which has contributed to the challenge of identifying consistent trends across locations. In this study, we introduce an innovative method for quantifying global SC intensity changes, irrespective of regional distinctions. Consequently, we identify a robust overall SC change across models. This robustness is rooted in the overall trend observed among models, despite the regional variations in SC alterations. To enhance clarity and novelty, we've revised the introduction and bolstered our study's rationale with additional references.

Additionally, we've expanded our experimentation, encompassing SSP5-8.5, abrupt-4xCO₂, CFMIP-3, SMILEs, and d4PDF (outlined in the Methods section). The results, including Fig. 2 and supplementary figures, corroborate the consistent decline in global SC. We've also examined SC changes in individual hemispheres (Supplementary Figs. S2 and S3).

As addressed in response to comment #1, our use of an LBM delves into the role of local diabatic heating. In the revised manuscript, we provide an in-depth exploration of mechanisms underlying the robust weakening of global SC.

We agree with the comment that the regional differences in SC changes are also important relative to the general changes in global SC. Although the regional change in SC is less discussed in this study, the newly developed method provides an efficient way to remove the overall changes to highlight the regional features of SC changes in further study. We have further refined this aspect in the Summary section (Lines 261–268).

3. There is a need for clarification on the statistical significance of the findings mentioned in lines 173-175.

Re: We have revised the sentence as “The rate of weakening in the low-level SC at the end of the abrupt-4xCO₂ experiment is about -1.14% (with a 5–95% range of -2.90% to 0.21%), per 1 K global surface warming.”

4. In the summary section, it is crucial to incorporate a discussion on how the results of this article differ or align with existing literature, particularly in cases where similar research has been previously conducted, particularly in the Southern Hemisphere.

Re: Thank you for the comment. To address this comment and related comments from other reviewers together, we extensively rewrote the introduction section to emphasize the novelty of this study in relation to prior literature. Some necessary references are also added.

In the revision, we further calculated the general intensity change in global SC in addition to the global SC changes month-by-month and latitude-by-latitude in the original manuscript. The overall change computation for global SC omits regional discrepancies, revealing the robust changes in global SC.

Response to reviewer #2:

Thank you for the insightful comments and detailed instruction on how to improve the manuscript. The quality of the manuscript has been greatly improved based on your comments. Our point-by-point reply follows, where the original comments are quoted in italic.

Major comments:

1. *“Robust changes”*. The authors have never defined how they determine “robust” in the SC changes. Is it based on a large model agreement, the significance of the change, the seasonal consistency, or any others? This is a piece of important information that the authors would need to include.

Re: Thank you for the constructive comment. Indeed, the SC changes vary across seasons and regions. The "robust" claim in our study relies on inter-model consistency in the global SC decrease. This contrasts with earlier research reporting divergent regional changes. Our revised introduction underscores this novelty and clarifies the specific facet of robustness.

Additionally, we enhance the intermodel significance test in alignment with the method commonly employed in IPCC AR6. For specifics on the robustness test, refer to the Method section (Lines 342–351), provided below for convenience:

*“**Test of robustness.** A change is considered to be robust if it meets two criteria that the ratio of multi-model mean to inter-model standard deviation of the change is greater than 1 and 66% of the models show a change greater than the internal-variability threshold γ . This test of robustness considers both the uncertainty across the models and from internal variability, which is similar to the approach used in IPCC AR6 WG1 report. The ratio test is stricter than the sign agreement test often used in previous studies to evaluate the inter-model robustness. The internal-variability threshold is defined as $\gamma = \sqrt{2} \cdot 1.645 \cdot \sigma_{30yr}$, where σ_{30yr} is the standard deviation of 30-year climatology of thirteen periods separated from 500-year simulation in the piControl in which the first 100 years are ignored.”*

2. *Abstract*. Overall, the abstract is too technical. Please rewrite it in a form that is more understandable to the public audience.

Re: We have revised the abstract thoroughly.

3. Introduction. The introduction is very unclear about what is still unknown in the SC changes, and what is new in this study compared to the previous literature. The introduction also lacks examples of how SC changes may impact the weather and climate. The motivation is unclear. Although the authors have cited some references, their discussion of these previous studies is not thorough enough. It is important to include a discussion in terms of the significance of this study. 4. Novelty. There have been a great number of studies using CMIP6 models to study SC changes, so what is the novelty of this study?

Re for #3 and #4:

We've revised the introduction section according to the suggestion, clarifying the motivation and novelty of this study.

Previous projections of a specific SC have shown considerable variability based on location and season, lacking consistent changes. Further examination demonstrates that SC change projections are sensitive to domain and metrics selection. This study introduces a novel metric, assessing overall global SC changes and latitude-by-latitude SC changes, and uncovers a robust decline in global SC due to greenhouse warming. Key findings include:

1. Development of a novel method to measure general global SC intensity change by mitigating regional disparities.
2. Identification of a robust decline in global SC, contrary to varied projections in earlier research.
3. Global scale projection indicates the increasing dominance of SST warming over direct CO₂ forcing, resulting in a consistent decrease.

These insights greatly contribute to comprehending subtropical climate changes in a warmer climate.

5. Method. This section does not provide enough information. For example, how the 34 models were selected. Are they based on the data availability or the model fidelity in simulating the SC? The baseline is the model would need to capture the SC well, as well as the ensemble members the authors used. Please discuss how many ensemble members are used in each model and what ensemble members the authors used. In addition, I am not sure why a short period is examined in this study. The historical run

is up to 2014 while the authors only used the runs up to 2000. I would encourage the authors to extend their period to reduce the uncertainty and have more robust results. Please also clarify why the historical runs are used from 1971. Most of the reanalysis data are from 1979. It is better to match the period between historical runs and reanalysis data when comparing the model fidelity in simulating the SC which is currently missing. I understand the authors want to use the multi-model mean for an easier interpretation of the results, it is still with significant scientific meaning that the authors would need to discuss the model spread. Especially as pointed out by the authors that there are large inconsistencies between the studies which imply a large model spread. This is one of the novelties that I can see from this study. In terms of the definition of the subtropical circulation, it is not sure why the center of subtropical circulation is calculated using a large latitude spread which includes the tropics. The authors would need to test the sensitivity of their results by adjusting this latitude range.

Re: Thank you for the careful suggestion.

In the revision, we have greatly enhanced the clarity of the method presentation. Our model selection of 34 models is only based on the availability of the CMIP6 dataset. We have incorporated the fidelity assessment of these models against the ERA5 dataset using a Taylor diagram (Supplementary Fig. 16). Most models exhibit strong performance in simulating subtropical circulation, leading us to maintain the full set of models for our current analyses.

Figure S16. Taylor diagram of monthly climatology of the 850 hPa streamfunction in the historical experiment for 34 CMIP6 model covering 60°S–60°N with respect to data from ECMWF Reanalysis v5. The time span is from 1979 to 2008.

In the revision, the period applied in most experiments is set to 1979–2008 following the suggestion, which does not influence the conclusions.

For the test of robustness across the models, we now include the inter-model spread as a metric. All the intensity changes in global SC from different experiments used in this study have been shown in one figure in the revised manuscript.

Moreover, we have introduced two additional sets of experiments, SMILEs and d4PDF, to assess the influence of internal variability and patterned SST warming uncertainty. Notably, these factors do not overshadow the projections.

Regarding the definition domain of SC, we have determined that the latitude range does not alter our conclusions. Therefore, in the revised manuscript, we have refined our scope to encompass 20°–40°N for the Northern Hemisphere and 20°–40°S for the Southern Hemisphere in calculating the center of subtropical circulation.

6. Readability. The authors want to make the manuscript concise, however, they lost clarity. Most of their description or statements are too general and hard to follow. Their figures are generally not straightforward, so it will be much more helpful if they could elaborate more in the main text. The mechanisms they discuss are also not clear and missing details. They would need to refine most of their main text to keep the manuscript concise while also making the statement clear and with enough details for the readers to understand.

Re: We agree with the comment regarding the omission of certain details in the initial manuscript, which was originally tailored for a different journal before its transfer to Nature Communications. Given the relatively flexible length constraints of Nature Communications, we have incorporated additional information in the revision. Specifically, the word count of the main text has been expanded from approximately 2100 to around 2800 words. Furthermore, the Method section has grown from about 235 words to approximately 1100 words. The inclusion of two more figures brings the total figure count to six, and supplementary materials have been enriched with supplementary information.

The key revisions encompass the following aspects:

1. A comprehensive rework of the introduction to elucidate the motivation and novelty of our study.
2. Incorporation of an analysis on the global SC's general intensity change.
3. Introduction of supplementary experiments aimed at assessing the impact of internal variability and the uncertainty associated with the pattern of SST change.
4. Integration of LBM (Large-scale Biogeochemical Model) experiments to explore the influence of SST forcing across distinct geographical regions.

All these supplementary experiments serve to further reinforce the robustness of our findings.

7. Figures. The stipplings are very hard to see in the figures. For Figure 2, please do not make -4 to 4 white. There is a large model agreement in the tropics but there is no color. Figure 3: The hatched pattern makes the figure messy and the stippling is rather hard to see. I think the authors could think of a better way to show the local weakening. Figure 4d: how this figure is calculated would need to be discussed in the Method section. And why 40S-40N is used by including the tropics?

Re: Apologies for any unclear figures; this may be due to the submission system's transformation impacting figure quality. In the revised version, we've made improvements: changing stippling to hatching, redrawing figures with suggested colormaps, and enhancing the presentation of local weakening.

We've also included a comprehensive explanation of the method for calculating intensity changes in global SC in the Methods section.

Regarding Figure 4d, we acknowledge that the previous domain selection (40S-40N) wasn't appropriate. In the updated manuscript, we compute the overall change in global SC and present new Figure 6a with a range of 10°–45°N for the Northern Hemisphere and 10°–45°S for the Southern Hemisphere.

To gauge how changes in the chosen domain's scope affect intensity change in global SC, we tested various ranges: 45°S–45°N, 10°–40°S and 10°–40°N, as well as 10°–50°S and 10°–50°N. The outcomes based on these different scopes (see Supplementary Fig. 17) exhibit minimal differences, indicating that the calculation of intensity change in global SC isn't significantly affected by the chosen domain's scope.

Figure S17. Intensity changes in the global subtropical circulation with respect to global-mean surface warming in SSP5-8.5. The intensity changes in the global subtropical circulation in SSP5-8.5 runs (years 2070–2099; 34 models) relative to historical runs (years 1979–2008) are calculation over 10°–45°S and 10°–45°N (black dots), 45°S–45°N (red dots), 10°–40°S and 10°–40°N (blue dots), and 10°–50°S and 10°–50°N (purple dots).

8. Discussion. A comprehensive discussion is necessary following the summary in terms of consistency with the previous studies and the implication we can get from this paper.

Re: We have revised the introduction thoroughly to clarify the motivation and novelty of this study and added two paragraphs in Summary discussing the implications (Lines 249–268).

The main implications include: a contracted monsoon range and drying at its boundary with arid regions, an importance of CO₂ emission trajectory and projection term, the major uncertainty sources, and the application of new method for regional SC changes.

9. Please provide a map showing the SST warming pattern in the supplementary. Or provide a more detailed description of the amip experiments such as the value of the warming, and the pattern of the warming.

Re: We have added Supplementary Fig. 7 to show the changes in the sea surface temperature and 850 hPa vector wind in the amip-future4K experiment relative to the amip-p4K experiment in each month.

10. Because the introduction is poorly written, it is hard to determine the significance and novelty of this manuscript. Please rewrite the introduction to show the novelty of this study. In addition, the manuscript is poorly written as I mentioned that some important details are missing, also the figures are hard to understand. Please try to include as many details in the Method section in terms of calculation and analysis methods. Most of the discussion cited mechanisms from previous papers and the main findings seemed to be the role of SST warming and CO₂ forcing. Therefore, this paper is more of a further analysis instead of a breakthrough finding. I would therefore recommend transferring this study to another journal. I would also encourage the authors to try to convert the manuscript into a longer paper to include more details and thorough discussion.

Re: We've extensively revised our manuscript to enhance clarity regarding the study's motivation and novelty. The key revisions encompass the following aspects:

1. A comprehensive rework of the introduction to elucidate the motivation and novelty of our study.
2. Incorporation of an analysis on the global SC's general intensity change.
3. Introduction of supplementary experiments aimed at assessing the impact of internal variability and the uncertainty associated with the pattern of SST change.
4. Integration of LBM (Large-scale Biogeochemical Model) experiments to explore the influence of SST forcing across distinct geographical regions.

Minor comments:

1. L6-8. L13-16. L18-20. Please rephrase, hard to understand.
2. L21-22. How do the results carry implications for subtropical climate changes?

Re. for #1 and #2: We have rewritten the abstract as in major comment #2. The main implications summarized in the abstract are: “The declined SC implies a contracted monsoon range and drying at its boundary with arid regions under CO₂-induced global warming, and the varied impact of processes across timescales highlights the essential role of assessing CO₂ emission trajectory and projection term.”

3. L33-34. *Is this something already found or the authors' hypothesis? Please clarify.*
4. L33. *“but no consistent response...” For example?*
5. L34-37. *Not linked to the previous sentence as to which mechanisms are in which seasons and which regions?*
6. L37-38. *It seems that it has been a well-established mechanism from the impact of the direct CO₂ effect. And this mechanism has been used to explain changes in SC during summer. So, what is new in this study?*
7. L41. *The mechanism is fairly unclear here.*
8. L43. *Same comment as above.*

Re. for #3 to #8: We have rewritten the introduction. Please see the revised introduction.

9. L48. *The authors have never explained why there is a need to examine the changes latitude-by-latitude.*
10. L50-52. *Please do not repeat this again. This has been discussed in the abstract.*
11. L53-57. *Please do not include a summary of the results here. The introduction should be more focused on the background of the field and the significance and novelty of this study.*

Re. for #9 to #11: We have rewritten this part (Lines 68–79) as follows:

“Here, we develop a novel method to extract intensity change in global SC independent of the domain and season selection. A robust weakening of global SC is projected under the Shared Socioeconomic Pathway (SSP) 5-8.5 scenario by the end of the 21st century in 34 CMIP6 models and in the abrupt-4xCO₂ experiments forced by an abrupt quadrupling of the pre-industrial CO₂ level in 32 CMIP6 models. The robustness of the weakening is verified by the single-model initial-condition large ensemble (SMILE) and the “Database for Policy Decision Making for Future Climate Change” (d4PDF) simulations, which consider the uncertainties from internal variability and SST warming pattern, respectively. The robust weakening of global SC is dominated by global-mean surface warming and partly counteracted by the direct CO₂ effect, confirmed by the third Phase of the Cloud Feedback Model Intercomparison Project (CFMIP-3) in CMIP6.”

12. *Need some description of what the SC changes are like and how they change in the figure when discussing Fig. 1.*

Re: We have added the description of what the SC changes are like in Lines 35–41:

“The projection for a regional SC is crucially dependent on the location and season (Fig. 1 and Supplementary Fig. 1). The SC changes is distinct during the boreal summer and

winter in the Northern Hemisphere, whereas the seasonal variation of the SC changes in the Southern Hemisphere is relatively weak. The typical seasonal SC changes include the weakened Australian summer monsoon (Fig. 1a), the almost unchanged East Asian summer monsoon, and the westward shifted North Atlantic subtropical high in JJA (Fig. 1b).”

13. L65-67. *How the changes in the tropical circulation contribute to the changes in SC. The authors need to clarify that.*

Re: We have clarified the contribution of changes in the tropical circulation to changes in SC in the second paragraph from the bottom of the section “Seasonal variation of the changes in the subtropical circulation” (Lines 185–189) as follows:

“The low-level SC weakening under uniform SST warming is spatially homogeneous throughout the subtropics, which is consistent with the decreased vertical velocity in the mid-level troposphere (Fig. 5c) and reflects the mechanism of the Sverdrup balance connected with the weakened tropical circulation.”

14. L65. *I do not see the tropical circulation change as seasonally consistent from Fig. 1a and 1b. They need better quantification.*

Re: This sentence has been removed from the revised manuscript.

15. L68-70. *This kind of discussion is good. The authors would need to have some similar discussion to indicate how the SC changes and its implication to the changes of the climate systems.*

Re: We have rewritten the summary section, adding two paragraphs to discuss the implications in Lines 249–268.

16. L71. *“and latitude”. I do not see this from the earlier discussion.*

Re: We have added an introduction about the change in SC in the revised manuscript. Here is the introduction added in Lines 35–36:

“The projection for a regional SC is crucially dependent on the location and season (Fig. 1 and Supplementary Fig. 1).”

17. L77-78. *The SC in the winter season is more of an indication of a weakening in the southward flank rather than a weakening of the center when comparing the center of SC and the center of SC changes.*

Re: We have revised this sentence in Lines 141–143 as:

“The Northern Hemisphere SC is weakened around the equatorward flank of the

climatological SC center throughout the year, except in JJA (Fig. 3).”

18. L85. *Briefly describe*

Re: This paragraph has been removed in the revised manuscript.

19. L90-91. *Briefly describe their differences.*

Re: The different impacts of the mechanisms have been shown in Fig. 2 in the revised manuscript. These mechanisms have also been introduced in the introduction.

20. L99-100. *Why this is specifically discussed? Unless this is important to some weather and climate systems.*

Re: This sentence has been removed in the revised manuscript.

21. L101. *Not always, only in a few months.*

Re: We have revised this sentence in Lines 149–151 as:

“This weakening shifts seasonally with the climatological SC center and occurs primarily on its equatorward flank in a few months.”

22. L104. *Please elaborate. The authors always make a general statement. They could state in more detail such as which latitudes and which seasons.*

23. L106-107. *Not a convincing mechanism here. The reduction in midlatitude baroclinicity is dominant in which seasons and caused by which SST change patterns? Please provide more details.*

Re for #22 and #23: We have revised here (Lines 152–160) as “The impact of the patterned SST warming (Supplementary Fig. 7) on the SC with the impact from uniform SST warming removed (Fig. 4b) is relatively weak compared to the uniform SST warming (Fig. 4a), but it enhances the poleward flank of the SC in the Southern Hemisphere during the austral summer (Fig. 4b and Supplementary Fig. 8). This enhancement is associated with the southeasterly wind changes over the southeastern Pacific (Supplementary Fig. 7), which is modulated by a stronger west-minus-east gradient during the austral summer around 20°S–40°S of the prescribed-SST change pattern (Supplementary Fig. 7).”

24. L138-139. *Can this mechanism explain why the strengthening most occurred in the poleward flank?*

Re: The strengthening most occurred in the poleward flank is explained by the paragraph (Lines 163–173) as follows:

“By contrast, direct CO₂ radiative forcing robustly strengthens the global SC, especially during the boreal summer monsoon season (June–September; Fig. 4c and Supplementary Fig. 10). Strengthening of the SC is stronger on the poleward flank of

the climatological SC center because of the superposition of the poleward shift in the midlatitude jet (Supplementary Fig. 11). Direct CO₂ radiative forcing can induce stratospheric cooling and a poleward shift of midlatitude jet throughout the year, favoring strengthening of the SC in both hemispheres. However, its impact is located around the midlatitude jets, exactly the poleward boundary of the climatological SC in both hemispheres (Fig. 4c and Supplementary Fig. 11), indicating the poleward shift in the midlatitude jet could not be the major factor enhancing SC, especially during the boreal summer.”

25. L355. Not straightforward how the seasonal changes are calculated. Needed a brief clarification here or in the main text.

Re: We have added a detailed introduction to how the intensity change in global SC and seasonal and latitudinal intensity change are calculated in the Methods.

26. L385. Where is the shading?

Re: The shading was lost due to unknown technical problems of the manuscript system. We have solved this issue in another way in current submission.

Response to reviewer #3:

Thank you for the insightful comments and detailed instructions on how to improve the manuscript. The quality of the manuscript has been greatly improved based on your comments. Our point-by-point reply follows, where the original comments are quoted in italics.

Major comments:

1. Significance: Many studies have already discussed the roles of global-mean warming, the patterns of SST changes, and direct CO₂ radiative forcing on the changes in the subtropical circulation as referred to in the manuscript. The main results of this study, such as the weakening of tropical circulation, the intensification of the Asian summer monsoon, and the poleward shift of SH subtropical circulation, confirm results from many previous studies. Thus, it seems that the main results are not novel and significant enough for Nature Communication.

Re: Thank you for the comments. We are sorry for some unclear representation in the original manuscript. In the revision, we have rewritten the introduction section to clarify the motivation and novelty of this study. Moreover, we have added a new section to emphasize the general decrease in global SC in the revision.

The main findings of this study include:

1. Development of a novel method to measure general global SC intensity change by mitigating regional disparities.
2. Identification of a robust decline in global SC, contrary to varied projections in earlier research.
3. Global scale projection indicates the increasing dominance of SST warming over direct CO₂ forcing, resulting in a consistent decrease.

These insights greatly contribute to comprehending subtropical climate changes in a warmer climate.

2. Validity/Data and Methodology: The description of the data and methodology is well provided. However, I think the comparison of AMIP-future4K and AIMP-p4K is not enough to address the role of the SST pattern changes. One of the dominant uncertainties in circulation changes is rooted in uncertainties in the future SST pattern changes (e.g., Chadwick et al., 2014 referred to in the manuscript). Since the AMIP-future4K uses the same mean SST pattern projected by CMIP simulations, the current

analysis has limitations in addressing the effect of future SST pattern changes. However, models tend to show robust responses to global-mean warming and direct CO₂ radiative forcing as discussed in many previous studies (e.g., Fahad et al., 2021; Bony et al., 2013; Vecchi et al., 2006; and many others referred to in this manuscript).

Re: Thank you for the constructive comments. We agree with the comment that the SST change pattern is an important source inducing great uncertainties in regional climate changes under global warming. This aspect was not adequately addressed in the original manuscript.

Following the suggestion, we conducted an additional analysis using a series of atmosphere-only experiments referred to as the Database for Policy Decision-Making for Future Climate Change (d4PDF). These experiments consist of six atmosphere-only simulations, each driven by a 4 K patterned warming derived from six distinct representative SST warming patterns (Supplementary Fig. 4). The corresponding findings are presented in Fig. 2, Supplementary Figs. 2, 3, 4, and 9.

The variability in intensity changes across the six d4PDF experiments remains within the range of inter-model variations observed in the SSP5-8.5 experiments. This outcome is consistent with a robust weakening of the global SC. Further elaboration on the outcomes of the d4PDF analyses can be found in Lines 98–104 and 160–162 of the manuscript.

Figure S4 Annual-mean changes in the surface air temperature in the **a**, HFB_4K_CC (CCSM4), **b**, HFB_4K_GF (GFDL-CM3), **c**, HFB_4K_HA (HadGEM2-AO), **d**, HFB_4K_MI (MIROC5), **e**, HFB_4K_MP (MPI-ESM-MR) and **f**, HFB_4K_MR (MRI-CGCM3) experiments relative to the HPB (present climate) experiment from the d4PDF. There are 15 members for each future experiments. Only the results of 15-member ensemble mean are shown at here. Because the SST warming pattern is not provided in the d4PDF, the change in surface air temperature is shown here as a reference for the SST forcing pattern.

3. *Clarity and Context: The manuscript is well written and clear. However, there are several limitations in context. First, the main results are only based on mean of multi-model changes. Fidelity of model simulation in the present and the role of model*

uncertainties in addressing future changes should be discussed. Second, the significant test is only based on the degree of the models agree on the sign of the multi-model mean in the study. Considering large internal variability in circulation changes, a significant test should be also applied with respect to internal variability.

Re: Thank you for the kind suggestion. Indeed, some certain details were omitted in the initial manuscript, which was originally tailored for a different journal before its transfer to Nature Communications. Given the relatively flexible length constraints of Nature Communications, we have incorporated additional information in the revision. Specifically, the word count of the main text has been expanded from approximately 2100 to around 2800 words. Furthermore, the Method section has grown from about 235 words to approximately 1100 words. The inclusion of two more figures brings the total figure count to six, and supplementary materials have been enriched with supplementary information.

The key revisions encompass the following aspects:

1. A comprehensive rework of the introduction to elucidate the motivation and novelty of our study.
2. Incorporation of an analysis on the global SC's general intensity change.
3. Introduction of supplementary experiments aimed at assessing the impact of internal variability and the uncertainty associated with the pattern of SST change.
4. Integration of LBM (Large-scale Biogeochemical Model) experiments to explore the influence of SST forcing across distinct geographical regions.

All these supplementary experiments serve to further reinforce the robustness of our findings.

Test of robustness:

In the revision, the definition of inter-model robustness across the models is revised to a more reasonable method in the revision, which is widely used in IPCC AR6. The details for the robustness test are shown in the Method section (Lines 342–351), provided below for convenience:

“Test of robustness. A change is considered to be robust if it meets two criteria that the ratio of multi-model mean to inter-model standard deviation of the change is greater than 1 and 66% of the models show a change greater than the internal-variability threshold γ . This test of robustness considers both the uncertainty across the models and from internal variability, which is similar to the approach used in IPCC AR6 WG1

report. The ratio test is stricter than the sign agreement test often used in previous studies to evaluate the inter-model robustness. The internal-variability threshold is defined as $\gamma = \sqrt{2} \cdot 1.645 \cdot \sigma_{30yr}$, where σ_{30yr} is the standard deviation of 30-year climatology of thirteen periods separated from 500-year simulation in the piControl in which the first 100 years are ignored.”

Model selection:

Our model selection of 34 models is only based on the availability of the CMIP6 dataset. We have incorporated the fidelity assessment of these models against the ERA5 dataset using a Taylor diagram (Supplementary Fig. 16). Most models exhibit strong performance in simulating subtropical circulation, leading us to maintain the full set of models for our current analyses.

Figure S16. Taylor diagram of monthly climatology of the 850 hPa streamfunction in the historical experiment for 34 CMIP6 model covering 60°S–60°N with respect to data from ECMWF Reanalysis v5. The time span is from 1979 to 2008.

Possible uncertainties from internal variability

We examined five initial-condition large ensembles from a single model to explore internal variability's potential influence. Figure 2 in the revised manuscript displays our findings. The internal variability's spread remains within the bounds of inter-model uncertainties.

REVIEWER COMMENTS

Reviewer #1 (Remarks to the Author):

no further comments from my end.

Reviewer #2 (Remarks to the Author):

The manuscript has largely been improved from its previous version by incorporating comments from all the reviewers. I appreciate all the effort and work the authors have done to strengthen their study. I am pleased with the revision and do not have any critical comments. I only have one suggestion is to further improve their introduction to emphasize the significance and novelty of this study. The novelty of a paper published in this journal cannot be just a novel method, only if that method can be widely applied. The authors would need to highlight what findings are new in their study compared to the previous literature. For example, is this the first study to compare the influence of SST warming and CO₂? Also, is the different timescales of these two impact factors new finding? And is the seasonal variation new? It would greatly enhance the novelty of this study if the authors could provide evidence that their findings are new.

Reviewer #3 (Remarks to the Author):

The revised manuscript has been substantially improved and has addressed most of my comments. However, some issues still require further clarification.

Major comments:

1. Representation of the subtropical circulation: This study uses the 850-hPa streamfunction to represent low-level subtropical circulation. Thus, this study may miss contributions from divergent circulation, which is also an important part of the monsoon system. A similar approach may apply to the 850-hPa velocity potential to check whether robust changes are also detected in the divergent circulation.
2. Definition of the intensity of Subtropical Circulation: I think the new definition for the intensity change in SC in this study is quite useful since it is independent on the domain and the seasonal evolution of subtropical circulation. However, I am concerned about calculating the intensity changes in global subtropical circulation. The ratio change may be too large over the region with a very small climatological value. Figure 1 indicates that large changes tend to occur over the edge region where sign changes. It would be useful to show a map of the intensity change (% shading) on top of the climatology of the 850-hPa streamfunction(contour) in the historical experiment as a supplementary figure (Similar to Figure 1

except for the intensity change with unit of %). The new figure will be useful to check the validity of the definition used in this study.

3. An expansion and shift of circulation: As discussed in the Summary section, the new definition of global subtropical circulation cannot address an expansion or shift of circulation, which may be more relevant to society. For example, the positive anomaly over the Arabian Sea, India, and the Bay of Bengal can be seen as a huge reduction in cyclonic circulation, but it can also be interpreted as a northward shift and intensification of the Indian Ocean anticyclonic circulation. This aspect may require more discussion in the Summary section.

Minor comments:

1. Lines 39~40: Figure 1 shows the weakening of the Australian summer monsoon and the strengthening of the East Asian summer monsoon (the intensification and eastward extension of the monsoon trough but the weakening of the western North Pacific anticyclonic circulation). However, both changes are not robust with respect to the inter-model spread. Interestingly, the text reads 'the weakened Australian summer monsoon' and the almost unchanged East Asia summer monsoon.' Should it be replaced with 'the slightly weakened Australian summer monsoon in DJF (Fig. 1a), the slightly strengthened East Asian summer monsoon and the robust westward shifted North Atlantic subtropical high in JJA (Fig. 1b)'?

2. Lines 98-101: It is good to see results from d4PDF. Although the six SST warming patterns are quite similar to each other, mostly having El Nino-like warming (Figure S4), the spread among SST patterns seems very large (e.g., Figure 2, Figure S2, and Figure S3). There are two extremes in the result: one is slight intensification, and the other is considerable weakening, as shown in Figure 2. The extreme SC intensification is especially considerable in the Southern Hemisphere, as shown in Figure S3. It may be good to check the two extremes and further discuss the role of the SST warming pattern on changes in subtropical circulation.

3. Figure 3: It may be good to add a line for the center of the subtropical circulation in the future.

Response to reviewer #2:

Thank you for the insightful comments and detailed instructions on how to improve the manuscript. The quality of the manuscript has been greatly improved based on your comments. Our point-by-point reply follows, where the original comments are quoted in italics.

1. The manuscript has largely been improved from its previous version by incorporating comments from all the reviewers. I appreciate all the effort and work the authors have done to strengthen their study. I am pleased with the revision and do not have any critical comments. I only have one suggestion is to further improve their introduction to emphasize the significance and novelty of this study. The novelty of a paper published in this journal cannot be just a novel method, only if that method can be widely applied. The authors would need to highlight what findings are new in their study compared to the previous literature. For example, is this the first study to compare the influence of SST warming and CO₂? Also, is the different timescales of these two impact factors new finding? And is the seasonal variation new? It would greatly enhance the novelty of this study if the authors could provide evidence that their findings are new.

Re: Thank you very much for recognizing our work and giving constructive suggestions to improve the presentation of the introduction further.

We have revised the last paragraph in the Introduction section to highlight our novelty in Lines 79–85.

Response to reviewer #3:

Thank you for the insightful comments and detailed instructions on how to improve the manuscript. The quality of the manuscript has been greatly improved based on your comments. Our point-by-point reply follows, where the original comments are quoted in italics.

The revised manuscript has been substantially improved and has addressed most of my comments. However, some issues still require further clarification.

Re: Thank you very much for recognizing our efforts. We have implemented all your additional comments, which further help us improve the results and presentation.

Major comments:

1. Representation of the subtropical circulation: This study uses the 850-hPa streamfunction to represent low-level subtropical circulation. Thus, this study may miss contributions from divergent circulation, which is also an important part of the monsoon system. A similar approach may apply to the 850-hPa velocity potential to check whether robust changes are also detected in the divergent circulation.

Re: Thank you very much for this reminder. Indeed, the divergent subtropical circulation is not dispensable, particularly during the summer monsoon seasons, although the rotational component is dominant in the subtropical circulation. Here, we only considered the rotational component mainly based on the progress of previous studies that reasonably demonstrated a weakening of the divergent SC due to a weakened tropical overturning circulation (Held and Soden 2006; Tanaka et al. 2004; Ueda et al. 2006; Vecchi and Soden 2007), whereas the change in the dominant rotational component of SC remains inconclusive (Cherchi et al. 2018; He et al. 2017; He and Zhou 2022; Li et al. 2013; Li et al. 2012; Shaw and Voigt 2015). As a result, most previous studies on the SC change focused on the dominant rotational component by analyzing the streamfunction of SC (He et al. 2017; Li et al. 2012; Shaw and Voigt 2015), as examined by default in our previous manuscript.

To clarify this point, we have revised the introduction section in Lines 35–38 as:

“When the divergent SC is projected to weaken due to a weakened tropical overturning circulation under global warming, the change in the dominant rotational component of

SC remains inconclusive. The projection for the rotational SC represented by 850 hPa streamfunction (hereafter referred to as SC)

Following your suggestion, we also confirm the earlier conclusion of a decline in the divergent SC resulting from the weakening of the tropical overturning circulation by analyzing the velocity potential at 850 hPa. As shown in Fig. R1 (also shown in Supplementary Fig. 4), the divergent component of SC demonstrates a more robust decrease in the SSP5-8.5 scenario, aligning with previous findings. The results support our present conclusion of a robust weakening in SC.

We have added a short paragraph in Lines 101–105 and a supplementary figure to discuss this point.

Figure R1 As in Fig. 2, but for the divergent component of subtropical circulation represented by the velocity potential at 850 hPa in the SSP5-8.5 runs.

References:

- Cherchi, A., T. Ambrizzi, S. Behera, A. C. V. Freitas, Y. Morioka, and T. Zhou, 2018: The response of subtropical highs to climate change. *Current Climate Change Reports*, 4, 371-382, <https://doi.org/10.1007/s40641-018-0114-1>.
- He, C., B. Wu, L. Zou, and T. Zhou, 2017: Responses of the summertime subtropical anticyclones to global warming. *J. Climate*, 30, 6465-6479, <https://doi.org/10.1175/jcli-d-16-0529.1>.
- He, C., and T. J. Zhou, 2022: Distinct responses of North Pacific and North Atlantic summertime subtropical anticyclones to global warming. *J. Climate*, 35, 4517-4532, <https://doi.org/10.1175/Jcli-D-21-1024.1>.
- Held, I. M., and B. J. Soden, 2006: Robust responses of the hydrological cycle to global warming. *J. Climate*, 19, 5686–5699, <https://doi.org/10.1175/jcli3990.1>.
- Li, W., L. Li, M. Ting, and Y. Liu, 2012: Intensification of northern hemisphere subtropical highs in a warming climate. *Nat. Geosci.*, 5, 830-834, <https://doi.org/10.1038/ngeo1590>.
- Li, W., and Coauthors, 2013: Intensification of the southern hemisphere summertime subtropical anticyclones in a warming climate. *Geophys. Res. Lett.*, 40, 5959-5964, <https://doi.org/10.1002/2013gl058124>.
- Shaw, T. A., and A. Voigt, 2015: Tug of war on summertime circulation between radiative forcing and sea surface warming. *Nat. Geosci.*, 8, 560-566, <https://doi.org/10.1038/ngeo2449>.
- Tanaka, H. L., N. Ishizaki, and A. Kitoh, 2004: Trend and interannual variability of walker, monsoon and Hadley circulations defined by velocity potential in the upper troposphere. *Tellus A: Dynamic Meteorology and Oceanography*, 56, <https://doi.org/10.3402/tellusa.v56i3.14410>.
- Ueda, H., A. Iwai, K. Kuwako, and M. E. Hori, 2006: Impact of anthropogenic forcing on the Asian summer monsoon as simulated by eight GCMs. *Geophys. Res. Lett.*, 33, L06703, <https://doi.org/10.1029/2005gl025336>.
- Vecchi, G. A., and B. J. Soden, 2007: Global warming and the weakening of the tropical circulation. *J. Climate*, 20, 4316–4340, <https://doi.org/10.1175/jcli4258.1>.

2. Definition of the intensity of Subtropical Circulation: I think the new definition for the intensity change in SC in this study is quite useful since it is independent on the domain and the seasonal evolution of subtropical circulation. However, I am concerned about calculating the intensity changes in global subtropical circulation. The ratio change may be too large over the region with a very small climatological value. Figure 1 indicates that large changes tend to occur over the edge region where sign changes. It would be useful to show a map of the intensity change (% , shading) on top of the climatology of the 850-hPa streamfunction(contour) in the historical experiment as a supplementary figure (Similar to Figure 1 except for the intensity change with unit of %). The new figure will be useful to check the validity of the definition used in this study.

Re: Thanks for your comments. Indeed, some centers of the SC changes represented by streamfunction are misaligned with those of the climatology. Moreover, the climatological SC streamfunction has sign changes. As a result, the ratio change relative to the climatology SC is not suitable for the girds with a climatology SC close to zero. That is the reason that previous studies had to select a relatively large domain to define the changes and/or the percentage. The projection method in this study defining the intensity change is developed to avoid this issue.

We have rephrased the “Intensity change” section in the Methods as follows:

“The projection method defining the percentage intensity change prevents unreasonable large values in cases where the climatological SC is close to 0, unlike the conventional method that divides changes by climatology.”

3. An expansion and shift of circulation: As discussed in the Summary section, the new definition of global subtropical circulation cannot address an expansion or shift of circulation, which may be more relevant to society. For example, the positive anomaly over the Arabian Sea, India, and the Bay of Bengal can be seen as a huge reduction in cyclonic circulation, but it can also be interpreted as a northward shift and intensification of the Indian Ocean anticyclonic circulation. This aspect may require more discussion in the Summary section.

Re: Following the suggestion, we have added more discussion about the limitations of our method in the Summary section as follows (Lines 284–288):

“Some shifts occur in regions where multiple subsystems of SC are interconnected,

making them difficult to interpret solely in terms of overall intensity change. Despite this, the new method can efficiently isolate overall intensity changes in global SC or in a relatively large domain, aiding the focus on more specific changes in regional SC in future research.”

Minor comments:

1. Lines 39~40: Figure 1 shows the weakening of the Australian summer monsoon and the strengthening of the East Asian summer monsoon (the intensification and eastward extension of the monsoon trough but the weakening of the western North Pacific anticyclonic circulation). However, both changes are not robust with respect to the inter-model spread. Interestingly, the text reads ‘the weakened Australian summer monsoon’ and the almost unchanged East Asia summer monsoon.’ Should it be replaced with ‘the slightly weakened Australian summer monsoon in DJF (Fig. 1a), the slightly strengthened East Asian summer monsoon and the robust westward shifted North Atlantic subtropical high in JJA (Fig. 1b)’?

Re: Corrected.

2. Lines 98-101: It is good to see results from d4PDF. Although the six SST warming patterns are quite similar to each other, mostly having El Nino-like warming (Figure S4), the spread among SST patterns seems very large (e.g., Figure 2, Figure S2, and Figure S3). There are two extremes in the result: one is slight intensification, and the other is considerable weakening, as shown in Figure 2. The extreme SC intensification is especially considerable in the Southern Hemisphere, as shown in Figure S3. It may be good to check the two extremes and further discuss the role of the SST warming pattern on changes in subtropical circulation.

Re: We have added more discussion on these two extreme results. The result of slightly intensified SC and considerably weakened SC are from HFB_4K_MI and HFB_4K_MR experiments, respectively. In HFB_4K_MI, a rare cooling occurs over the Southern Ocean, enhancing the meridional temperature gradient and strengthening westerlies (Ceppi et al. 2018). Additionally, there is a relatively weak El Niño-like warming in the tropical Pacific, resulting in a weaker westerly change over the equatorial western Pacific. Both these features in HFB_4K_MI contribute to strengthening the Southern Hemisphere SC.

The discussions are added in Lines 123–131 as follows:

“We contrast two extreme results in the six d4PDF experiments: one involving slight intensification and the other a considerable weakening (Fig. 2), particularly in the Southern Hemisphere (Supplementary Figs. 2 and 3). These results are associated with HFB_4K_MI (Supplementary Fig. 5d) and HFB_4K_MR (Supplementary Fig. 5f), respectively. In HFB_4K_MI, a rare cooling occurs over the Southern Ocean, enhancing the meridional temperature gradient and strengthening westerlies. Additionally, there is a relatively weak El Niño-like warming in the tropical Pacific, resulting in a weaker westerly change over the equatorial western Pacific. Both these features in HFB_4K_MI contribute to strengthening the Southern Hemisphere SC.”

References:

Ceppi, P., G. Zappa, T. G. Shepherd, and J. M. Gregory, 2018: Fast and slow components of the extratropical atmospheric circulation response to CO₂ forcing. *J. Climate*, 31, 1091–1105, <https://doi.org/10.1175/jcli-d-17-0323.1>.

3. *Figure 3: It may be good to add a line for the center of the subtropical circulation in the future.*

Re: Dashed curves for the future centers of the subtropical circulation have been added in Fig. 3.

REVIEWERS' COMMENTS

Reviewer #3 (Remarks to the Author):

The authors have clarified my concerns and addressed my comments well. I recommend to accept the manuscript for publication.